# Flowtron: an Autoregressive Flow-based Generative Network for Text-to-Speech Synthesis

**Rafael Valle, Kevin J. Shih, Ryan Prenger & Bryan Catanzaro**
NVIDIA
`rafaelvalle@nvidia.com`

## Abstract

In this paper we propose Flowtron: an autoregressive flow-based generative network for text-to-speech synthesis with style transfer and speech variation. Flowtron borrows insights from Autoregressive Flows and revamps Tacotron 2 in order to provide high-quality and expressive mel-spectrogram synthesis. Flowtron is optimized by maximizing the likelihood of the training data, which makes training simple and stable. Flowtron learns an invertible mapping of data to a latent space that can be used to modulate many aspects of speech synthesis (timbre, expressivity, accent). Our mean opinion scores (MOS) show that Flowtron matches state-of-the-art TTS models in terms of speech quality. We provide results on speech variation, interpolation over time between samples and style transfer between seen and unseen speakers. Code and pre-trained models are publicly available at https://github.com/NVIDIA/flowtron.

## 1 Introduction

Current speech synthesis methods do not give the user enough control over how speech actually sounds. Automatically converting text to audio that successfully communicates the text was achieved a long time ago (Umeda et al., 1968; Badham et al., 1983). However, communicating only the text information leaves out the acoustic properties of the voice that convey much of the meaning and human expressiveness. In spite of this, the typical speech synthesis problem is formulated as a text to speech (TTS) problem in which the user inputs only text since the 1960s. This work proposes a normalizing flow model (Kingma & Dhariwal, 2018; Huang et al., 2018) that learns an unsupervised mapping from non-textual information to manipulable latent Gaussian distributions.

Taming the non-textual information in speech is difficult because the non-textual is unlabeled. A voice actor may speak the same text with different emphasis or emotion based on context, but it is unclear how to label a particular reading. Without labels for the non-textual information, recent approaches (Shen et al., 2017; Arik et al., 2017a;b; Ping et al., 2017) have formulated speech synthesis as a TTS problem wherein the non-textual information is implicitly learned. Despite their success in recreating non-textual information in the training set, the user has limited insight and control over it.

It is possible to formulate an unsupervised learning problem in such a way that the user can exploit the unlabeled characteristics of a data set. One way is to formulate the problem such that the data is assumed to have a representation in some latent space, and have the model learn that representation. This latent space can then be investigated and manipulated to give the user more control over the generative model's output. Such approaches have been popular in image generation, allowing users to interpolate smoothly between images and to identify portions of the latent space that correlate with various features (Radford et al., 2015; Kingma & Dhariwal, 2018; Izmailov et al., 2019).

Recent deep learning approaches to expressive speech synthesis have combined text and learned latent embeddings for non-textual information (Wang et al., 2018; Skerry-Ryan et al., 2018; Hsu et al., 2018; Habib et al., 2019; Sun et al., 2020). These approaches impose an undesirable paradox: they require making assumptions before hand about the dimensionality of the embeddings when the correct dimensionality can only be determined after the model is trained. Even then, these embeddings are not guaranteed to contain all the non-textual information it takes to reconstruct speech, often

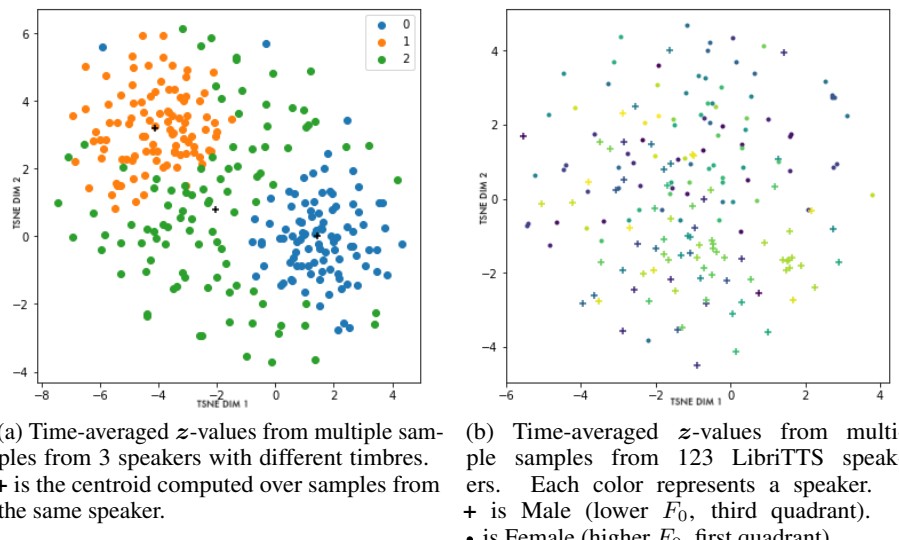

(a) Time-averaged $z$-values from multiple samples from 3 speakers with different timbres. **+** is the centroid computed over samples from the same speaker.

(b) Time-averaged $z$-values from multiple samples from 123 LibriTTS speakers. Each color represents a speaker. **+** is Male (lower $F_0$, third quadrant). **•** is Female (higher $F_0$, first quadrant).

Figure 1: T-SNE plot showing Flowtron partitioning the $z$-space according to acoustic characteristics.

resulting in models with dummy or uninterpretable latent dimensions and not enough capacity, as the appendices in Wang et al. (2018); Skerry-Ryan et al. (2018); Hsu et al. (2018) confirm.

Furthermore, most models are not able to manipulate speech characteristics over time due to fixed-length embeddings. Their assumption is that variable-length embeddings are not robust to text and speaker perturbations (Skerry-Ryan et al., 2018), which we show not to be the case. Finally, although VAEs and GANs (Sun et al., 2020; Habib et al., 2019; Hsu et al., 2018; Bińkowski et al., 2019; Akuzawa et al., 2018) provide a latent embedding that can be manipulated, they may be difficult to train, are limited to *approximate* latent variable prediction, and rely on an implicit generative model or ELBO estimate to perform MLE in the latent space (Kingma & Dhariwal, 2018; Lucic et al., 2018; Kingma et al., 2016).

In this paper we propose Flowtron: an autoregressive flow-based generative network for mel-spectrogram synthesis with style transfer over time and speech variation. Flowtron learns an invertible function that maps a distribution over mel-spectrograms to a latent $z$-space parameterized by a spherical Gaussian. Figure 1 shows that acoustic characteristics like timbre and $F_0$ correlate with portions of the $z$-space of Flowtron models trained without speaker embeddings.

With our formalization, we can generate samples containing specific speech characteristics manifested in mel-space by finding and sampling the corresponding region in $z$-space (Gambardella et al., 2019). Our formulation also allows us to impose a structure to the $z$-space and to parametrize it with a Gaussian mixture, similar to Hsu et al. (2018). In our simplest setup, we generate samples with a zero mean spherical Gaussian prior and control the amount of variation by adjusting its variance.

Compared to VAEs and GANs and their disadvantages enumerated in Kingma & Dhariwal (2018), manipulating a latent prior in Flowtron comes at no cost in speech quality nor optimization challenges. Flowtron is able to generalize and produce sharp mel-spectrograms, even at high $\sigma^2$ values, by simply maximizing the likelihood of the data while not requiring any additional Prenet or Postnet layer (Wang et al., 2017), nor compound loss functions required by most SOTA models (Shen et al., 2017; Ping et al., 2017; Skerry-Ryan et al., 2018; Wang et al., 2018; Bińkowski et al., 2019).

In summary, Flowtron is optimized by maximizing the exact likelihood of the training data, which makes training simple and stable. Using normalizing flows, it learns an invertible mapping from data to latent space that can be manipulated to modulate many aspects of speech synthesis. Concurrent with this work are Glow-TTS (Kim et al., 2020) and Flow-TTS (Miao et al., 2020), both of which incorporate normalizing flows into the TTS task. Our work differs from these two in that Flowtron is an autoregressive architecture where we explore the use of flow to modulate speech and style variation. In contrast, Glow-TTS and Flow-TTS are parallel architectures that focus on inference

speed. Our mean opinion scores (MOS) show that Flowtron matches SOTA TTS models in terms of speech quality. Further, we provide results on speech variation, interpolation between samples and interpolation between styles over time, and style transfer between seen and unseen speakers with equal or different sentences. We hope this work, the first to show evidence that normalizing flows can be used for *expressive* text-to-speech synthesis and style transfer, will further stimulate developments in normalizing flows.

## 2 FLOWTRON

Flowtron is an autoregressive flow that generates a sequence of mel-spectrogram frames. A normalizing flow generates samples by first sampling a latent variable from a known distribution $p(\boldsymbol{z})$, and applying a series of invertible transformations to produce a sample from the target distribution $p(\boldsymbol{x})$. These invertible transformations $\boldsymbol{f}$ are known as steps of flow:

$$\boldsymbol{x} = \boldsymbol{f}_1 \circ \boldsymbol{f}_2 \circ \ldots \boldsymbol{f}_k(\boldsymbol{z}) \tag{1}$$

Because each transformation is invertible, we can directly evaluate the exact log-likelihood of the target distribution $p(\boldsymbol{x})$ using the change of variables:

$$\log p_\theta(\boldsymbol{x}) = \log p_\theta(\boldsymbol{z}) + \sum_{i=1}^{k} \log |\det(\boldsymbol{J}(\boldsymbol{f}_i^{-1}(\boldsymbol{x})))| \tag{2}$$

$$\boldsymbol{z} = \boldsymbol{f}_k^{-1} \circ \boldsymbol{f}_{k-1}^{-1} \circ \ldots \boldsymbol{f}_1^{-1}(\boldsymbol{x}) \tag{3}$$

Where $\boldsymbol{J}$ is the Jacobian of the inverse transform $\boldsymbol{f}_i^{-1}(\boldsymbol{x})$. By cleverly choosing the latent distribution $p(\boldsymbol{z})$ and the invertible transformations, the exact log-likelihood becomes simple and tractable.

### 2.1 LATENT DISTRIBUTIONS

We consider two simple distributions for the latent distribution $\boldsymbol{z}$: a zero-mean spherical Gaussian and a mixture of spherical Gaussians with fixed or learnable parameters.

$$\boldsymbol{z} \sim \mathcal{N}(\boldsymbol{z}; 0, \boldsymbol{I}) \quad \text{or} \quad \boldsymbol{z} \sim \sum_k \hat{\phi}_k \, \mathcal{N}(\boldsymbol{z}; \hat{\boldsymbol{\mu}}_k, \hat{\boldsymbol{\Sigma}}_k) \tag{4}$$

The zero-mean spherical Gaussian has a simple log-likelihood. The mixture of the spherical Gaussians, has inherent clusters that might result in interesting aspects of the audio information.

### 2.2 INVERTIBLE TRANSFORMATIONS

Normalizing flows are typically constructed using coupling layers (Dinh et al., 2014; 2016; Kingma & Dhariwal, 2018). In our case, we use an autoregressive affine coupling layer (Dinh et al., 2016). The latent variable $\boldsymbol{z}$ has the same number of dimensions and frames as the resulting mel-spectrogram sample. The previous frames $\boldsymbol{z}_{1:t-1}$ produce scale and bias terms, $\boldsymbol{s}_t$ and $\boldsymbol{b}_t$ respectively, that affine-transform the succeeding time step $\boldsymbol{z}_t$:

$$(\log \boldsymbol{s}_t, \boldsymbol{b}_t) = NN(\boldsymbol{z}_{1:t-1}, \boldsymbol{text}, \boldsymbol{speaker}) \tag{5}$$

$$\boldsymbol{f}(\boldsymbol{z}_t) = (\boldsymbol{z}_t - \boldsymbol{b}_t) \div \boldsymbol{s}_t \tag{6}$$

$$\boldsymbol{f}^{-1}(\boldsymbol{z}_t) = \boldsymbol{s}_t \odot \boldsymbol{z}_t + \boldsymbol{b}_t \tag{7}$$

Here, $NN()$ can be any autoregressive causal transformation (Shumway & Stoffer, 2017). The affine coupling layer is a reversible transformation, even though $NN()$ itself need not be invertible. We use a 0-vector for obtaining the scaling and bias terms what will affine transform $\boldsymbol{z}_1$. This 0-vector constant also guarantees that the first $\boldsymbol{z}$ is always known.

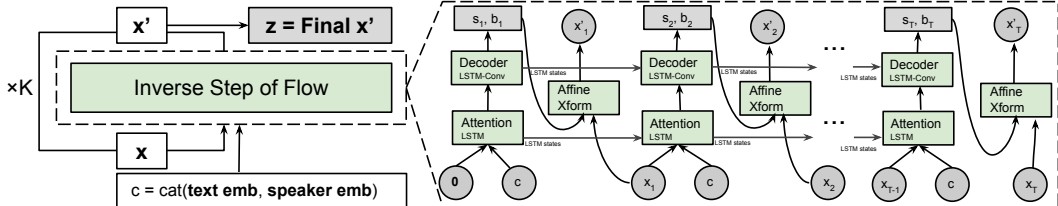

Figure 2: Mapping data ($\mathbf{x}$) to the latent dimension ($\mathbf{z}$) for K steps of flow. The right side shows a unrolled view of the $NN()$ architecture. Text and speaker embeddings are channel-wise concatenated to produce context matrix $c$. At every time step $t$, a recurrent attention mechanism computes a weighting distribution over the context matrix $c$ to produce a weighted-sum reduction over $c$, which is then passed through an LSTM-Conv decoder architecture to generate affine parameters for transforming $x_{t+1} \rightarrow x'_{t+1}$. As predicted parameters are always for the *next* time step, the first iteration is conditioned on a pre-defined 0-vector.

With an affine coupling layer, only the $\boldsymbol{s_t}$ term changes the volume of the mapping and adds a change of variables term to the loss. This term also penalizes the model for non-invertible affine mappings.

$$\log |\det(\boldsymbol{J}(\boldsymbol{f}^{-1}_{coupling}(\boldsymbol{x})))| = \log |\boldsymbol{s}| \tag{8}$$

To evaluate the likelihood, we take the mel-spectrograms and pass them through the inverse steps of flow conditioned on the text and optional speaker ids, adding the corresponding $\log |\boldsymbol{s}|$ penalties, and evaluate the result based on the Gaussian likelihoods.

With this setup, it is also possible to reverse the ordering of the mel-spectrogram frames in time without loss of generality. We reverse the order of frames on even steps of flow, defining a step of flow as a full pass over the input sequence. This allows the model to learn dependencies both forward and backwards in time while remaining causal and invertible.

### 2.3 Model architecture

Our text encoder modifies the text encoder in Tacotron 2 by replacing batch-norm with instance-norm. Our decoder and $NN$ architecture, depicted in Figure 2, removes the Prenet and Postnet layers from Tacotron previously thought to be essential (Shen et al., 2017). Please compare Figure 2 describing our architecture and Figure 8 in A.4.4 describing Tacotron's architecture. We also provide model summary views in A.6

We use the content-based tanh attention described in Vinyals et al. (2015), which can be easily modified to become also location sensitive. We use the Mel Encoder described in Hsu et al. (2018) to predict the parameters of the Gaussian Mixture. Following (Valle et al., 2019b), we use speaker-embeddings channel-wise concatenated with the encoder outputs at every token. We use a single shared embedding for models not conditioned on speaker id.

The step of flow closest to the latent variable $\boldsymbol{z}$ has a gating mechanism that prunes extra frames from the $\boldsymbol{z}$-values provided to the model during inference. The length of $\boldsymbol{z}$-values remains fixed on the next steps of flow.

### 2.4 Inference

Inference, given a trained model, is simply a matter of sampling $\boldsymbol{z}$ values from a spherical Gaussian, or Gaussian Mixture, and running them through the network in the forward direction $f$, e.g. Eq. 1. The parameters of the Gaussian mixture are either fixed or predicted by Flowtron. Training was conducted with $\sigma^2 = 1$, but we explore the effects of different values for $\sigma^2$ in section 3.3. In general, we found that sampling $\boldsymbol{z}$ from a Gaussian with lower standard deviation than used during training resulted in better sounding mel-spectrograms, as similarly concluded in Kingma & Dhariwal (2018) and (Parmar et al., 2018). Our inference results use $\sigma^2 = 0.5$ while sampling the prior and the posterior variance while sampling the posterior.

## 2.5 POSTERIOR INFERENCE

Figure 1 shows that several speech characteristics present in mel-spectrograms are clustered into regions of the $z$-space. Knowing this, we can treat the latent distribution as a prior $q(z) = \mathcal{N}(0, I)$ and obtain a posterior over the latent space of the flow model $q(z|\zeta_{1:m})$ conditioned on the evidence $\zeta_{1:m}$, which are $m$ data observations $x_i$ mapped to the latent space using $\zeta_i = f^{-1}(x_i)$. We can use a Gaussian likelihood function with covariance matrix $\Sigma$ to compute the posterior above analytically, $q(z|\zeta_{1:m}) = \mathcal{N}(\mu_p, \Sigma_p)$. Following the approach in Gambardella et al. (2019), defining $\bar{\zeta}$ as the mean of $\zeta_i$ and using $\lambda$ as a hyperparameter, we define the parameters of the posterior below. Please see A.2, Algorithm 1 and Gambardella et al. (2019) for implementation details and a full derivation.

$$\boldsymbol{\mu}_p = \frac{\frac{m}{\lambda}\bar{\zeta}}{\frac{m}{\lambda} + 1} \quad \boldsymbol{\Sigma}_p = \frac{1}{\frac{m}{\lambda} + 1}I \tag{9}$$

## 3 EXPERIMENTS

This section describes our training setup and provides quantitative and qualitative results. Our quantitative results show that Flowtron has mean opinion scores that are comparable to the state of the art. Our qualitative results demonstrate many features that are either impossible or inefficient to achieve using Tacotron, Tacotron 2 GST and Tacotron GM-VAE. These features include variation control in speech, interpolation between samples, and style transfer over time.

We decode all mel-spectrograms into waveforms with a WaveGlow (Prenger et al., 2019) model available on github (Valle et al., 2019a). This suggests that WaveGlow can be used as an universal decoder. In addition to our illustrated and quantitative results, we ask that the readers listen to Flowtron samples in our supplementary materials corresponding to our qualitative experiments.

### 3.1 TRAINING SETUP

We train Flowtron, Tacotron 2 and Tacotron 2 GST models using a dataset (LSH) that combines the LJSpeech dataset (Ito et al., 2017) with two proprietary single speaker datasets with 20 and 10 hours each (Sally and Helen). We also train a Flowtron model on the *train-clean-100* subset of LibriTTS (Zen et al., 2019) with 123 speakers and 25 minutes on average per speaker. Speakers with less than 5 minutes of data and files that are larger than 10 seconds are filtered out. For each dataset, we use at least 180 samples for the validation set, and the remainder for the training set.

The models are trained on uniformly sampled normalized text and ARPAbet encodings obtained from the CMU Pronouncing Dictionary (Weide, 1998). We do not perform any data augmentation. We adapt public Tacotron 2 and Tacotron 2 GST repos to include speaker embeddings as described in Section 2. We use the same mel-spectrogram representation used in WaveGlow (Prenger et al., 2019). We train Flowtron with a pre-trained text encoder, progressively adding steps of flow once the last step of flow has learned to attend to text. Flowtron models used in our experiments have 2 steps of flow. We forward readers to A.3 and A.4 for details on our training setup and ablation studies.

### 3.2 MEAN OPINION SCORE (MOS) COMPARISON

We use the LJS voice as a reference and compare MOS between real samples, samples from Flowtron with 2 steps of flow, and samples from Tacotron 2. Following guidelines in (Prenger et al., 2019), we crowd-sourced MOS tests on Amazon Mechanical Turk using 30 volume normalized utterances disjoint from the training set for evaluation, and randomly chose the utterances for each subject. The scores provided in (Prenger et al., 2019) are used for real samples.

The mean opinion scores are shown in Table 1 with 95% confidence intervals computed over approximately 250 scores per source. The results roughly match our subjective qualitative assessment. The larger advantage of Flowtron is in the control over the amount of speech variation and the manipulation of the latent space.

| Source | Flows | MOS |
|---|---|---|
| Real | - | $4.27 \pm 0.13$ |
| Flowtron | 2 | $3.66 \pm 0.16$ |
| Tacotron 2 | - | $3.52 \pm 0.17$ |

Table 1: Mean Opinion Scores

### 3.3 SAMPLING THE PRIOR

The simplest approach to generating samples
with Flowtron is to sample from a prior distribution $z \sim \mathcal{N}(0, \sigma^2)$ and adjust $\sigma^2$ to control the amount of variation. Whereas $\sigma^2 = 0$ completely removes variation and produces outputs based on the model bias, increasing $\sigma^2$ will increase the amount of variation in speech.

#### 3.3.1 SPEECH VARIATION

We illustrate the relationship between $\sigma^2$ and control over variability by synthesizing Flowtron samples with $\sigma^2 \in \{0.0, 0.5, 1.0\}$. All samples are generated conditioned on the speaker *Sally* and the text "*How much variation is there?*". Despite the variability added by increasing $\sigma^2$, all Flowtron-synthesized samples produce high quality speech.

Figure 3 shows that contrary to commonly held wisdom (Shen et al., 2017; Arik et al., 2017a;b; Ping et al., 2017; Skerry-Ryan et al., 2018; Wang et al., 2018; Bińkowski et al., 2019), Flowtron generates sharp harmonics and well resolved formants without a compound loss nor Prenet or Postnet layers.

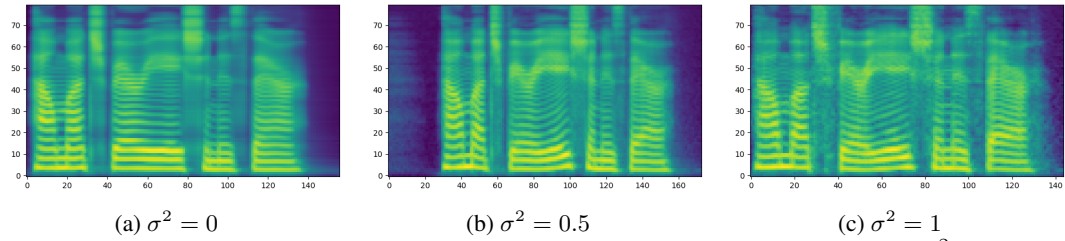

(a) $\sigma^2 = 0$        (b) $\sigma^2 = 0.5$        (c) $\sigma^2 = 1$

Figure 3: Flowtron Mel-spectrograms illustrate increasing variability by using different $\sigma^2$ and that Flowtron is able to produce sharp harmonics with high $\sigma^2$ and without Prenet or Postnet layers.

Now we show that adjusting $\sigma^2$ is a simple and valuable approach that provides more variation and control thereof than Tacotron, without sacrificing speech quality and despite of having a similar but simpler architecture. For this, we synthesize 10 samples with Tacotron 2 using different values for the Prenet dropout probability $p \in \{0.45, 0.5, 0.55\}$, scaling outputs accordingly. Samples computed on values of $p \in [0.3, 0.8]$ are not included because they sound unintelligible.

Figure 4 provides plots of $F_0$ contours extracted with the YIN algorithm (De Cheveigné & Kawahara, 2002), with minimum $F_0$, maximum $F_0$, and harmonicity threshold equal to 80 Hz, 400 Hz and 0.3 respectively. Our results are similar to the previous sample duration analysis. As expected, $\sigma^2 = 0$ provides no variation in $F_0$ contour[1], while increasing $\sigma^2$ will increase variation in $F_0$ contours.

Our results in Figure 4 also show that Flowtron samples are considerably less monotonous than the samples produced with Tacotron 2 at no cost and with a similar but simpler architecture. Whereas increasing $\sigma^2$ considerably increases variation in $F_0$, modifying $p$ barely produces any variation. This is valuable because expressive speech is associated with non-monotonic $F_0$ contours. In A.1 we show similar results with respect to sentence duration.

#### 3.3.2 INTERPOLATION BETWEEN SAMPLES

With Flowtron, we can perform interpolation in $z$-space to achieve interpolation in mel-spectrogram space. This experiment evaluates Flowtron models with and without speaker embeddings. For the experiment with speaker embeddings, we choose the speaker Sally and the phrase "*It is well known that deep generative models have a rich latent space.*". We generate mel-spectrograms by sampling $z \sim \mathcal{N}(0, 0.8)$ twice and interpolating between them over 100 timesteps. For the experiment without speaker embeddings we interpolate between Sally and Helen using the phrase "*We are testing this model.*".

First, we perform inference by sampling $z \sim \mathcal{N}(0, 0.5)$ until we find $z$ values, $z_h$ and $z_s$, that produce mel-spectrograms with Helen's and Sally's voice respectively. We then generate samples by

---

[1]Variations in $\sigma^2 = 0$ are due to different $z$ for WaveGlow.

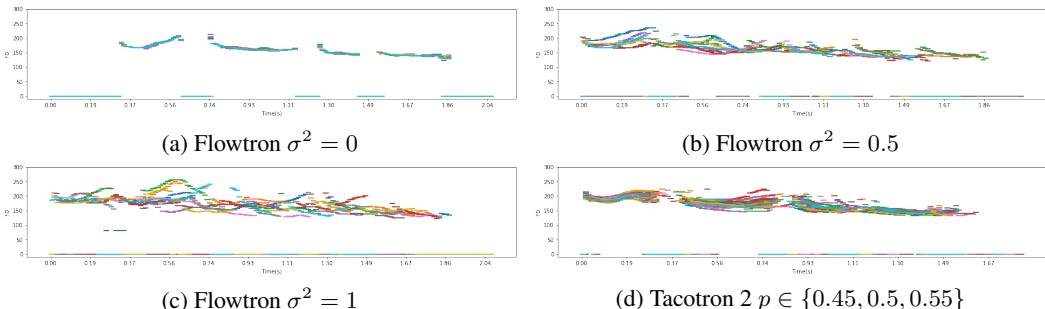

(a) Flowtron $\sigma^2 = 0$                          (b) Flowtron $\sigma^2 = 0.5$

(c) Flowtron $\sigma^2 = 1$                          (d) Tacotron 2 $p \in \{0.45, 0.5, 0.55\}$

Figure 4: $F_0$ contours obtained from samples generated by Flowtron and Tacotron 2 with different values for $\sigma^2$ and $p$. Flowtron provides more variability and expressivity than Tacotron 2.

performing inference while linearly interpolating between $z_h$ and $z_s$. Our same speaker interpolation samples show that Flowtron is able to interpolate between multiple samples while producing correct alignment maps. Our different speaker interpolation samples show that Flowtron is able to gradually and smoothly morph one voice into another.

### 3.4 SAMPLING THE POSTERIOR (STYLE TRANSFER)

We generate samples with Flowtron by sampling a posterior distribution conditioned on the evidence containing speech characteristics of interest, as described in 2.5 and Gambardella et al. (2019). Tacotron 2 GST Wang et al. (2018) has an equivalent posterior sampling approach. During inference, the model is conditioned on a weighted sum of global style tokens (posterior) queried through an embedding of existing audio samples (evidence). We evaluate Tacotron 2 GST using a single sample to query a style token, or multiple samples to compute an average style token. For complete results, please refer to audio samples in the supplemental material corresponding to the following sections.

#### 3.4.1 SEEN SPEAKER

In this section we run two style transfer experiments: the first one (Expressive) uses samples with high variance in pitch, which we use as a proxy for comparing expressivity in speech; the second (High Pitch), uses samples with high average pitch. In these experiments, we provide comparisons between Pitch Mean and Pitch Standard Deviation from the *Reference* samples providing the style, a Flowtron Baseline and after style transfer using Flowtron Posterior and Tacotron 2 GST.

Our experiments show that by sampling from the posterior or interpolating between the posterior and the Gaussian prior over time, Flowtron makes a monotonic speaker gradually sound more expressive. Architectures similar to Tacotron 2 GST with fixed-latent embeddings are not able to perform gradual changes in style over time. Table 2 provides pitch summary statistics computed over 5 phrases and 10 takes each and shows that Flowtron is overall closer to the reference providing the style than Tacotron 2 GST. Our supplemental materials also show that Tacotron 2 GST sentences are repetitive and contain vocal-fry like distortions.

| Model \ Style | Pitch Mean | | | Pitch Standard Deviation | | |
|---|---|---|---|---|---|---|
| | Expressive | High Pitch | Surprised | Expressive | High Pitch | Surprised |
| *Reference* | 53.6 | 55.2 | 58.2 | 4.5 | 2.3 | 1.0 |
| FTA Posterior | **53.4** | 53.3 | **55.5** | **2.5** | **2.3** | 3.0 |
| FTA Baseline | 53.1 | 52.6 | 52.8 | 2.2 | 1.9 | 1.9 |
| Tacotron 2 GST | 51.7 | **53.6** | 51.6 | 2.0 | 2.4 | **1.7** |

Table 2: Values closer to the *Reference* are better. Comparison between pitch (MIDI number) summary statistics from *Reference* providing the style, Flowtron with standard Gaussian prior (FTA Baseline) and samples after style transfer with Flowtron (FTA Posterior) and Tacotron 2 GST. Our results show that FTA Posterior is overall more effective than Tacotron 2 GST in emulating the *Reference* by better matching its pitch summary statistics.

### 3.4.2 SEEN SPEAKER WITH UNSEEN STYLE

We compare samples generated with Flowtron and Tacotron 2 GST to evaluate their ability to emulate a speaking style unseen during training of a speaker seen during training. While Sally's data used during training consists of news article readings, the evaluation samples contain Sally's interpretation of the somber and vampiresque novel, *Born of Darkness* (BOD).

Our samples show that while Tacotron 2 GST fails to emulate the somber timbre in Born of Darkness, Flowtron succeeds in transferring not only the somber timbre, but also the low $F_0$ and the long pauses associated with the narrative style.

### 3.4.3 UNSEEN SPEAKER

In this experiment we compare Flowtron and Tacotron 2 GST samples to evaluate their ability to emulate the speaking style of a speaker not seen during training. The styles comes from speaker ID 24 and her "surprised" samples in RAVDESS (Livingstone & Russo, 2018), a dataset with emotion labels. Table 2 shows that while the samples generated with Tacotron 2 GST are not able to emulate the high-pitched style from RAVDESS, Flowtron is able to make Sally sound high-pitched as in the "surprised" style.

### 3.5 INTERPOLATION BETWEEN STYLES (PRIOR AND POSTERIOR)

In this experiment we illustrate how to control the speaking style at inference time by adjusting the parameter $\lambda$ in Equation 9 to interpolate between a baseline style (prior) and a target style (posterior). We use a model trained on LibriTTS and use a single sample from Sally's (unseen speaker) *Born of Darkness* dataset as evidence providing the target style. We synthesize posterior samples generated with Flowtron with $\lambda \in \{0.1, 0.666, 1.0, 2.0\}$. Figure 5 reflects the interpolation in style as interpolation in spectral profiles. Our supplemental materials aurally reflect a similar interpolation in other non-textual characteristics.

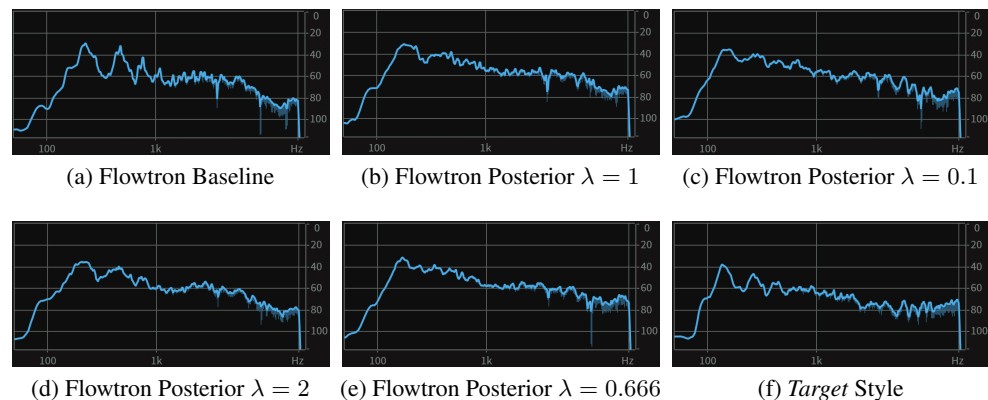

| (a) Flowtron Baseline | (b) Flowtron Posterior $\lambda = 1$ | (c) Flowtron Posterior $\lambda = 0.1$ |
|---|---|---|
| (d) Flowtron Posterior $\lambda = 2$ | (e) Flowtron Posterior $\lambda = 0.666$ | (f) *Target* Style |

Figure 5: Spectral profiles from the *target* style, from the Flowtron baseline generated using the prior, and from Flowtron samples generated using the posterior with different values for $\lambda$. These images show that by decreasing the value of $\lambda$ we gradually move the spectral profile from the baseline style (prior) to the target style (posterior).

### 3.6 SAMPLING THE GAUSSIAN MIXTURE

In this last section we provide samples from Flowtron Gaussian Mixture (GM) and visualizations. We replicate the experiments in Tacotron GM-VAE (Hsu et al., 2018) to visualize how speakers are assigned to mixture components and provide samples in which we modulate speech characteristics by translating one of the dimensions of an individual mixture component.

For these experiments, Flowtron GM-LibriTTS is trained on LibriTTS without speaker embeddings and a Gaussian mixture with 8 component with predicted mean, covariance and component assignment

probabilities; Flowtron GM-LSH is trained on LSH with speaker embeddings and a Gaussian Mixture with 8 components, fixed mean and covariances and predicted component assignment probabilities.

### 3.6.1 VISUALIZING ASSIGNMENTS

We evaluate model interpretability on a subset of LibriTTS with 123 speakers and 1410 utterances, 180 of which come from the validation set. Following Hsu et al. (2018), each utterance is assigned to the component with the highest posterior probability $\arg \max_k p(\hat{\phi}_k \mid \mathbf{x})$. We obtain posterior probabilities per utterance by using the Mel Encoder described in Section 2.3 and averaging the predicted component assignment probabilities over time. Figure 6 suggests that information in each component of Flowtron GM-LibriTTS is gender dependent.

We quantify the association between gender and mixture components with the metric described in Hsu et al. (2018). The assignment consistency with respect to gender is defined as $\frac{1}{M} \sum_{i=1}^{N} \sum_{j=1}^{N_i} \mathbb{1} y_{ij} = \hat{y}_i$, where $M$ is the number of utterances, $y_{ij}$ is the component assignment of utterance $j$ from speaker $i$, and $\hat{y}_i$ is the mode of $\{y_{ij}\}_{j=1}^{N_i}$. The assignment consistency in Flowtron GM-LibriTTS is $82.4\%$, suggesting that the components group utterances by speaker and group speakers by gender. We provide visualizations in Figure 6.

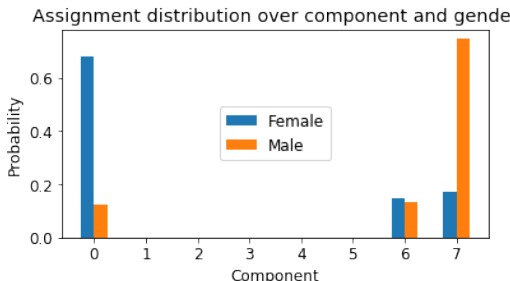

Figure 6: Component assignments suggest that information in each component is gender dependent.

### 3.6.2 TRANSLATING DIMENSIONS

We use the model Flowtron GM-LSH and focus on translating one of the dimensions of a single mixture component by adding an offset. The samples in our supplementary material show that we are able to modulate specific speech characteristics like pitch and word duration. Although the samples generated by translating one the dimensions associated with pitch height have different pitch contours, they have the same duration. Similarly, our samples show that translating the dimension associated with length of the first word does not modulate the pitch of the first word. We provide visualizations in Figure 9 in A.5.

## 4 CONCLUSION

We propose a new text to mel-spectrogram synthesis model based on autoregressive flows that is optimized by maximizing the likelihood and allows for speech variation and style transfer. Our results show that samples generated with Flowtron achieve mean opinion scores similar to SOTA TTS models. We demonstrate that our model learns a latent space that stores non-textual information, supervised using only MLE. Flowtron is able to produce high quality speech with high variability by adjusting $\sigma^2$.

Our results show that the latent space over non-textual features that can be investigated and manipulated to give the user more control over the generative model's output. We provide many examples that showcase this, including transferring the style from speakers seen and unseen during training to another speaker using sentences with similar or different text, and making a monotonic speaker sound more expressive. For future work, we are interested in using normalizing flows for few-shot speech synthesis, speech compression and in semi-supervised settings to exploit datasets with limited labels.

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

# A    Appendix

## A.1    Speech Variation

Figure 7 provides plots from sample durations in seconds. Our results show that larger values of $\sigma^2$ produces samples with more variation in duration, whereas $\sigma^2 = 0$ is fully deterministic. These results demonstrate that our latent space is able to model duration, which is a critical non-textual component to expressiveness in speech.

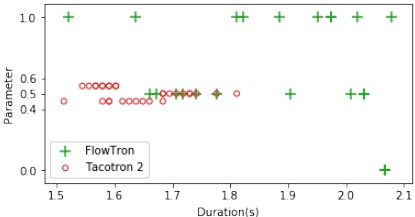

Figure 7: Sample duration given $\sigma^2$ and $p$ show that Flowtron provides more variation in sample duration than Tacotron.

## A.2    Posterior Inference

We generate posterior samples with Flowtron by sampling a posterior distribution conditioned on evidence containing speech characteristics of interest, as described in (Gambardella et al., 2019). We collect the evidence by performing a forward pass with Flowtron using with a speaker embedding, ($s \sim \mathcal{N}(0, I)$), the observed mel-spectrogram, and the text from a set of samples with the speech characteristics of interest. We use a specific speaker embedding when we want to factor out information about a specific speaker from $\zeta$.

Next, we compute $\bar{\zeta}$ by averaging $\zeta_{i,k}$ over batch $(i)$ or over batch and time $(i, k)$ and use Equation 9 to compute the parameters of the posterior. When averaging over batch, we repeat the z-values over the time dimension until they reach the desired length. We find in our experiments that averaging over batch is more efficient for transfering the style than averaging over batch and time. In all experiments, we select the best performing samples given $\lambda$ values between $m * 0.1$ and $m * 4$, where $m$ is the number of samples in the evidence. While small $\lambda$ values move the mean of the posterior closer to the evidence and decreases its variance, large $\lambda$ values move the mean of the posterior closer to the prior and increase the variance.

Once the parameters of the posterior distribution are computed, we can sample the posterior distribution and perform inference with the desired text and speaker. Algorithm 1 provides a description of posterior inference with Flowtron.

## A.3    Training Details

We use the ADAM (Kingma & Ba, 2014) optimizer with default parameters, 1e-4 learning rate and1e-6 weight decay for Flowtron and 1e-3 learning rate and 1e-5 weight decay for the other models,following Wang et al. (2017). We anneal the learning rate once the generalization error starts toplateau and stop training once the the generalization error stops significantly decreasing or startsincreasing. Flowtron models with 2 steps of flow were trained on the LSH dataset for approximately1000 epochs, then fine-tuned on LibriTTS for 500 epochs. Tacotron 2 and Tacotron 2 GST are trained for approximately 500 epochs. Each model is trained on a single NVIDIA DGX-1 with 8 GPUs.

---

**Algorithm 1:** Flowtron Posterior inference

---

**Input :** Trained Flowtron model $\boldsymbol{f}$, evidence audio samples $\boldsymbol{x}_{1:m}$
**Output :** Posterior sample

1 For each $mel_{i,k}, text_i, speaker_i \in \boldsymbol{x}_{1:m}$
  $\zeta_{i,k} \leftarrow \boldsymbol{f}^{-1}(mel_{i,k}, text_i, speaker_i)$

2 **if** *average over batch* **then**
3      repeat each $\zeta_k$ over the time dimension until target length is achieved
4      $\bar{\zeta}_k \leftarrow$ Compute $\zeta_{i,k}$ average over batch $k$
5 **else**
6      $\bar{\zeta} \leftarrow$ Compute $\zeta_{i,k}$ average over batch and time $k$
7 **end**
8 $\boldsymbol{\mu}_p, \boldsymbol{\Sigma}_p \leftarrow$ Compute posterior parameters using Equation 9
9 Initialize $\boldsymbol{Z}_p \sim \mathcal{N}(\boldsymbol{\mu}_p, \boldsymbol{\Sigma}_p)$
10 Sample $\boldsymbol{z}_p$ from $\boldsymbol{Z}_p$
11 Perform inference with Flowtron using $\boldsymbol{z}_p$, $text$ and $speaker$

---

### A.4 ABLATION STUDIES

#### A.4.1 COMPOSING FLOWS

We evaluated Flowtron models with 2, 3 and 6 steps of flow and found that more steps of flow have better likelihood but no significant qualitative improvement, while increasing inference time significantly. Hence, we chose to report results on Flowtron models with 2 steps of flow.

#### A.4.2 BIDIRECTIONAL PROCESSING

We compared the bidirectional (reversing the ordering of the mel-spectrogram frames in time on even numbered steps of flows) and unidirectional processing and found that bidirectional processing provides better likelihood and audio quality. Hence, we use bidirectional processing in all our Flowtron models.

#### A.4.3 ADDITIVE VS AFFINE TRANSFORMATIONS

The Tacotron 2 baseline without the postnet layer can be interpreted as additive single step autoregressive normalizing flow (ASSANF). By comparing Flowtron with Tacotron 2, we're comparing with a model that is better than an (ASSANF), as Tacotron 2 *sans* Postnet does not have sharp harmonics. Hence, we prefer affine over additive transformations.

#### A.4.4 COMPARISON WITH TACOTRON 2

The Tacotron 2 baseline without the postnet layer can be interpreted as additive single step autoregressive normalizing flow (ASSANF). By comparing Flowtron with Tacotron 2, we're comparing with a model that is better than an (ASSANF), as Tacotron 2 *sans* Postnet does not have sharp harmonics. Hence, we prefer affine over additive transformations.

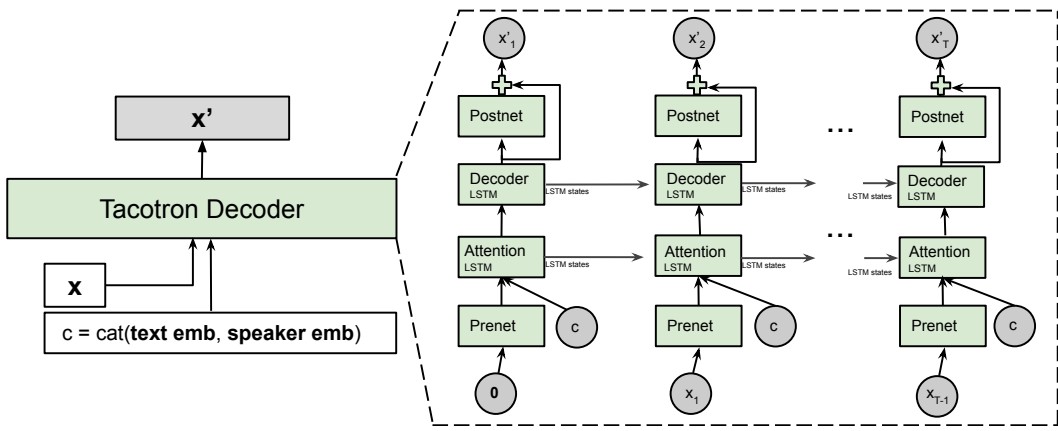

Figure 8: Visualization of the decoder in Tacotron 2 during training. Unlike Flowtron, Tacotron 2 requires Prenet and Postnet layers to learn attention and produce sharp harmonics.

## A.5 TRANSLATING DIMENSIONS

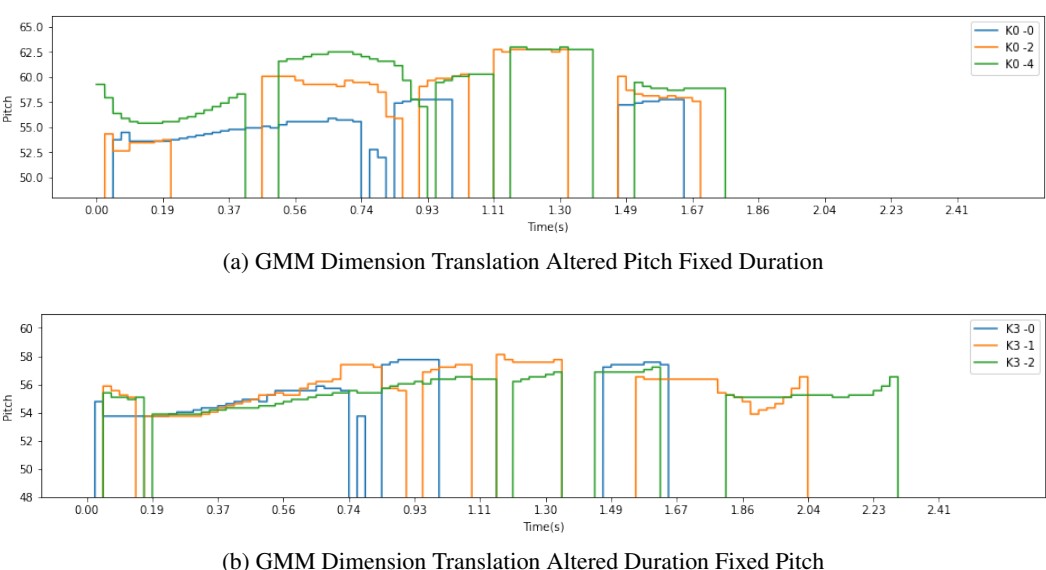

(a) GMM Dimension Translation Altered Pitch Fixed Duration

(b) GMM Dimension Translation Altered Duration Fixed Pitch

Figure 9: (a) shows that by translating one of the dimensions of $z$ we are able to alter the pitch contour of the sentence while keeping the length fixed. (b) shows that by translation one of the dimensions of $z$ we are able to alter the length of the sentence while keeping a similar pitch contour.

## A.6 FLOWTRON AND TACOTRON SUMMARY VIEW

```
Flowtron(
  (speaker_embedding): Embedding(3, 128)
  (embedding): Embedding(185, 512)
  (flows): ModuleList(
    (0): AR_Step(
      (conv): Conv1d(1024, 160, kernel_size=(1,), stride=(1,))
      (lstm): LSTM(1664, 1024, num_layers=2)
      (attention_lstm): LSTM(80, 1024)
      (attention_layer): Attention(
        (softmax): Softmax(dim=2)
        (query): LinearNorm(
          (linear_layer): Linear(in_features=1024, out_features=640, bias=False)
        )
        (key): LinearNorm(
          (linear_layer): Linear(in_features=640, out_features=640, bias=False)
        )
        (value): LinearNorm(
          (linear_layer): Linear(in_features=640, out_features=640, bias=False)
        )
        (v): LinearNorm(
          (linear_layer): Linear(in_features=640, out_features=1, bias=False)
        )
      )
      (dense_layer): DenseLayer(
        (layers): ModuleList(
          (0): LinearNorm(
            (linear_layer): Linear(in_features=1024, out_features=1024, bias=True)
          )
          (1): LinearNorm(
            (linear_layer): Linear(in_features=1024, out_features=1024, bias=True)
          )
        )
      )
    )
    (1): AR_Back_Step(
      (ar_step): AR_Step(
        (conv): Conv1d(1024, 160, kernel_size=(1,), stride=(1,))
        (lstm): LSTM(1664, 1024, num_layers=2)
        (attention_lstm): LSTM(80, 1024)
        (attention_layer): Attention(
          (softmax): Softmax(dim=2)
          (query): LinearNorm(
            (linear_layer): Linear(in_features=1024, out_features=640, bias=False)
          )
          (key): LinearNorm(
            (linear_layer): Linear(in_features=640, out_features=640, bias=False)
          )
          (value): LinearNorm(
            (linear_layer): Linear(in_features=640, out_features=640, bias=False)
          )
          (v): LinearNorm(
            (linear_layer): Linear(in_features=640, out_features=1, bias=False)
          )
        )
        (dense_layer): DenseLayer(
          (layers): ModuleList(
            (0): LinearNorm(
              (linear_layer): Linear(in_features=1024, out_features=1024, bias=True)
            )
            (1): LinearNorm(
              (linear_layer): Linear(in_features=1024, out_features=1024, bias=True)
            )
          )
        )
        (gate_layer): LinearNorm(
          (linear_layer): Linear(in_features=1664, out_features=1, bias=True)
        )
      )
    )
  )
  (encoder): Encoder(
    (convolutions): ModuleList(
      (0): Sequential(
        (0): ConvNorm(
          (conv): Conv1d(512, 512, kernel_size=(5,), stride=(1,), padding=(2,))
        )
        (1): InstanceNorm1d(512, eps=1e-05, momentum=0.1, affine=True, track_running_stats=False)
      )
```

```
      (1): Sequential(
        (0): ConvNorm(
          (conv): Conv1d(512, 512, kernel_size=(5,), stride=(1,), padding=(2,))
        )
        (1): InstanceNorm1d(512, eps=1e-05, momentum=0.1, affine=True, track_running_stats=False)
      )
      (2): Sequential(
        (0): ConvNorm(
          (conv): Conv1d(512, 512, kernel_size=(5,), stride=(1,), padding=(2,))
        )
        (1): InstanceNorm1d(512, eps=1e-05, momentum=0.1, affine=True, track_running_stats=False)
      )
    )
    (lstm): LSTM(512, 256, batch_first=True, bidirectional=True)
  )
)

Tacotron2(
  (embedding): Embedding(185, 512)
  (encoder): Encoder(
    (convolutions): ModuleList(
      (0): Sequential(
        (0): ConvNorm(
          (conv): Conv1d(512, 512, kernel_size=(5,), stride=(1,), padding=(2,))
        )
        (1): BatchNorm1d(512, eps=1e-05, momentum=0.1, affine=True, track_running_stats=True)
      )
      (1): Sequential(
        (0): ConvNorm(
          (conv): Conv1d(512, 512, kernel_size=(5,), stride=(1,), padding=(2,))
        )
        (1): BatchNorm1d(512, eps=1e-05, momentum=0.1, affine=True, track_running_stats=True)
      )
      (2): Sequential(
        (0): ConvNorm(
          (conv): Conv1d(512, 512, kernel_size=(5,), stride=(1,), padding=(2,))
        )
        (1): BatchNorm1d(512, eps=1e-05, momentum=0.1, affine=True, track_running_stats=True)
      )
    )
    (lstm): LSTM(512, 256, batch_first=True, bidirectional=True)
  )
  (decoder): Decoder(
    (prenet): Prenet(
      (layers): ModuleList(
        (0): LinearNorm(
          (linear_layer): Linear(in_features=80, out_features=256, bias=False)
        )
        (1): LinearNorm(
          (linear_layer): Linear(in_features=256, out_features=256, bias=False)
        )
      )
    )
    (attention_rnn): LSTMCell(896, 1024)
    (attention_layer): Attention(
      (query_layer): LinearNorm(
        (linear_layer): Linear(in_features=1024, out_features=128, bias=False)
      )
      (memory_layer): LinearNorm(
        (linear_layer): Linear(in_features=640, out_features=128, bias=False)
      )
      (v): LinearNorm(
        (linear_layer): Linear(in_features=128, out_features=1, bias=False)
      )
      (location_layer): LocationLayer(
        (location_conv): ConvNorm(
          (conv): Conv1d(2, 32, kernel_size=(31,), stride=(1,), padding=(15,), bias=False)
        )
        (location_dense): LinearNorm(
          (linear_layer): Linear(in_features=32, out_features=128, bias=False)
        )
      )
    )
    (decoder_rnn): LSTMCell(1664, 1024, bias=1)
    (linear_projection): LinearNorm(
      (linear_layer): Linear(in_features=1664, out_features=80, bias=True)
    )
    (gate_layer): LinearNorm(
      (linear_layer): Linear(in_features=1664, out_features=1, bias=True)
    )
```

```
  )
  (postnet): Postnet(
    (convolutions): ModuleList(
      (0): Sequential(
        (0): ConvNorm(
          (conv): Conv1d(80, 512, kernel_size=(5,), stride=(1,), padding=(2,))
        )
        (1): BatchNorm1d(512, eps=1e-05, momentum=0.1, affine=wTrue, track_running_stats=True)
      )
      (1): Sequential(
        (0): ConvNorm(
          (conv): Conv1d(512, 512, kernel_size=(5,), stride=(1,), padding=(2,))
        )
        (1): BatchNorm1d(512, eps=1e-05, momentum=0.1, affine=True, track_running_stats=True)
      )
      (2): Sequential(
        (0): ConvNorm(
          (conv): Conv1d(512, 512, kernel_size=(5,), stride=(1,), padding=(2,))
        )
        (1): BatchNorm1d(512, eps=1e-05, momentum=0.1, affine=True, track_running_stats=True)
      )
      (3): Sequential(
        (0): ConvNorm(
          (conv): Conv1d(512, 512, kernel_size=(5,), stride=(1,), padding=(2,))
        )
        (1): BatchNorm1d(512, eps=1e-05, momentum=0.1, affine=True, track_running_stats=True)
      )
      (4): Sequential(
        (0): ConvNorm(
          (conv): Conv1d(512, 80, kernel_size=(5,), stride=(1,), padding=(2,))
        )
        (1): BatchNorm1d(80, eps=1e-05, momentum=0.1, affine=True, track_running_stats=True)
      )
    )
  )
  (speaker_embedding): Embedding(3, 128)
)
```

