# OpenReview forum: "Flowtron: an Autoregressive Flow-based Generative Network for Text-to-Speech Synthesis"
_ICLR.cc/2021/Conference — ICLR 2021 Poster_

### Official Review · AnonReviewer3 · 2020-10-28
**Accept. The idea is good and experiments are good. There are some concerns about the clarity of the paper but those can be worked on.**

**Rating:** 9
**Confidence:** 5

**Review:**

### Summary

This paper introduces a novel application of normalizing flows to speech synthesis, allowing direct optimization of spectrogram log-likelihoods which results in more natural variation at inference compared to L1/L2 losses that model the mean. This setup also allows more control over non-textual information and interpolation between samples and styles.


### Recommendation

**Accept**
The idea is good and experiments are good. There are some concerns about the clarity of the paper but those can be worked on.

### Positives

1. The paper introduces a novel architecture and demonstrates improved output variation and more controllability, which is an important current issue for TTS research.
1. Many experiments investigating controllability are described, as well as some ablation studies for the model architecture in the appendix.


### Negatives

1. **Figure 1**: Please provide more informative captions. In the text, timbre and F0 are mentioned but it is not clear how that is relevant in the images. It is not clear what the colors mean in 1b, and if 1b is supposed to show a separation between male and female speakers the colors make it worse. Finally, are the points in 1b cluster centers or a single random sample per speaker?

1. **Figure 2**: Would it be possible to make it clearer that there is an autoregressive dependency in the Attention and Decoder blocks due to the LSTM cell memory? The way the figure is currently drawn makes it seem as if each attention/decoder block can be computed in parallel from only the inputs from the previous flow iteration.

1. In section 2.2 NN and f are described as acting on the latent variable frames $z_t$, but Figure 2 applies the NN and flow on the mel spectrogram frames x in order to produce z. Similarly, there is text saying "we take the mel-spectrograms and pass them through the inverse steps of flow" and "Inference, [...], is simply a matter of sampling z values [...] and running them through the network" but Figure 2 marks the block as "Step of Flow" rather than "Inverse Step of Flow". It would be good to smooth out the consistency.

1. More discussion about the end of sequence prediction would be appreciated. As the entire sequence of z must be used for inference, I assume there is some constant max inference length z used to obtain a final x, and the end of sequence prediction only happens to the final x rather than at each step of the flow? How well does this model adapt to inference samples that are much longer than any of the inputs seen during training?

1. **3.3.2 Interpolation Between Samples**: I don't understand why there was a need to sample z to find z_h and z_s. Is it not possible to take z_h and z_s from a random training example for the speaker? Or does that mean there is a correlation between the latent space z and the __content__ that is being spoken? If the latter, I would like to see more discussion on that.

1. **Table 2**: I assume bold means closest to ground truth? It's really hard to know how to interpret this table and I don't think it supports saying that FTA Posterior is more effective. FTA did not capture the increase in pitch mean from expressive -> high pitch, nor the decrease in std from expressive -> surprised. A better visualization may be to express this table as a bar graph (3 separate groups of 3 bars each) and conclude FTA Posterior is able to produce much more variation than Tacotron 2 GST?

### Misc

#### Abstract

1. "varation" -> "variation"
1. spell out IAF
1. "We provide results on speech variation" etc. sounds weak. eg. "Flowtron produces output with far more natural variation compared with Tacotron 2 and enables interpolation over time between samples and style transfer between seen and unseen speakers in ways that are either impossible or inefficient to achieve with prior works."

#### 1 Introduction

1. "Their assumption is that variable-length embeddings are not robust to text and speaker perturbations" citation needed
1. "Flowtron learns an invertible function that maps a distribution over mel-spectrograms to a latent z-space parameterized by a spherical Gaussian." mention IAF and cite Kingma et al., 2016 here instead of the previous paragraph.
1. "Finally, although VAEs and GANs provide a latent embedding that can be manipulated, they may be difficult to train, are limited to approximate latent variable prediction" While the IAF approach allows for direct optimization of log-likelihood, the latent variable encoding part is still approximate, just like in a VAE. The original Kingma paper even states that it is an approximate posterior.

#### 2.2 Invertible Transformations

1. $f^{-1}$ wouldn't be applied to $z_t$. Maybe $f^{-1}(x_t)$ or $f^{-1}\left(f(z_t)\right)$.

#### 3.1. Training Setup

1. "progressively adding steps of flow on the last step of flow has learned to attend to text" on->once?

#### 3.4.2 Seen Speaker with Unseen Style

1. "Flowtron succeeds in transferring not only the somber timbre, the low F0 and the long pauses
associated with the narrative style" -> "not only the somber timbre, but also"

#### Appendix

1. IAF is known to be quite inefficient. Is there any noticeable impacts on training or inference time? Or is this not a problem due to only using 2 layers of flow? It would be great if the ablation study in the appendix regarding layers of flow also covers this, but this is quite optional.

---

> ### Author Response · Authors · 2020-11-18
> **Response to AnonReviewer3 11/17/2020**
>
> Thank you for your kind words and thoughtful review. It brings us joy when a reviewer with high confidence asks us to change our text from "We provide results on speech variation" to "Flowtron produces output with far more natural variation compared with Tacotron 2 and enables interpolation over time between samples and style transfer between seen and unseen speakers in ways that are either impossible or inefficient to achieve with prior works."
>
> We have altered the paper to address the points you have brought in your review. Please take a look at the modifications in the paper and the clarifications below and let us know what remains to be addressed.
>
> Regarding Figure 1:
> We altered Figure’s 1 caption to clarify. Different colors represent different speakers, + is used for male speakers and dot is used for female speakers.
>
> Regarding Figure 2:
> Thank you for the suggestion, we added the LSTM dependencies to Figure 2.
>
> Regarding step of flow and inverse step of flow nomenclature.
> Thank you for your calling our attention to this. We consolidated the nomenclature for consistency.
>
> Regarding end of sequence prediction:
> The step of flow closest to the latent variable has a gating mechanism that prunes extra frames from the z-value provided to the model during inference. The resulting z-length after pruning is then fixed for the subsequent steps of flow. Hence, not necessarily the entire sequence of z will be used during inference, e. g. imagine passing 10 second worth of random z-values to synthesize the phrase “This is good.” We’ve clarified this in the paper.
>
> Regarding adapting to samples that are longer than inputs during training:
> Similar to Tacotron and other models with attention, the attention mechanism in Flowtron can fail on sentences that are longer than the inputs seen during training. With loss of generality, more robust attention mechanisms such as “Location-Relative Attention Mechanisms For Robust Long-Form Speech Synthesis” by Battenberg. et al. can be used to replace Flowtron's attention mechanism. Although not in the scope of this paper, in the near future we will add such modifications to our repo.
>
> Regarding interpolation between samples:
> Theoretically there should be no correlation between the conditions, i.e. text input, and the z-values. Empirically, our posterior inference samples show evidence that we can sample the posterior while using text different from the evidence used to compute the posterior. Hence, we should be able to do what you suggest. Simply put: we did not think about it...
>
> Regarding Table 2:
> Bold means closes to the reference sample providing the style. Hence, the goal of samples generated with style transfer is to match pitch summary statistics from reference samples providing the style. We've edited the text and the caption for Table 2 to clarify. We included a couple of cases in which Flowtron does not perform as well as Tacotron 2 GST to show possible areas for improvement.
>
> Regarding Variable Length Embeddings:
> We've added the citation to "Towards End-to-End prosody" by RJ Skerry-Ryan et al. and quote the paper here:  “In our experiments, variable-length prosody embeddings are able to generalize to very long utterances; however, compared to fixed-length embeddings, variable-length embeddings are not as robust to text and speaker perturbations likely because they encode a stronger timing signal. Therefore, this paper focuses on fixed-length embeddings.”
>
> Regarding Inverse Autoregressive Flows and Approximate Posteriors:
> The setup in the “Improved Variational Inference with IAF” uses an IAF to compute an approximate posterior q(z|x) for variational inference. Exact latent-variable inference and log-likelihood evaluation is a property of normalizing flows models in general, including IAFs. Also, we will clarify that even though our inspiration to write Flowtron came from IAFs and the Parallel Wavenet paper, Flowtron is an Autoregressive flow, also known as Masked Autoregressive Flow.
>
> Regarding the appendix and model performance:
> In our setup there are temporal dependencies in time both in training and inference due to the LSTM states, like you pointed out. Flowtron with 2 steps of flow is 7 times faster than real-time while Tacotron 2 is 10 times faster than real-time.

---

> > ### Comment · AnonReviewer3 · 2020-11-24
> > **Re: Response to AnonReviewer3 11/17/2020**
> >
> > Thank you for the reply. The updated paper looks great, the only nitpick I have is about the whitespace on top of page 6 and whether there is a mismatch between caption and subgraph in Figure 5. As the concerns have been addressed, I am updating the rating to 9 as I believe this paper is top 15%.

---

### Official Review · AnonReviewer2 · 2020-10-28
**Proposes a flow-based TTS with high expressivity and style variation.**

**Rating:** 5
**Confidence:** 3

**Review:**

This paper presents a text-to-speech synthesis system, called Flowtron which uses a normalizing flow to generate a sequence of mel-spectrogram frames. The difference between the proposed Flowtron and the previously prosed flow-based methods is, the authors argue as the main contributions, its ability to produce more diverse and expressive speech samples of specific speech attributes by sampling from the latent space. Evaluation is done using the two public datasets, and a number of experiments are performed to show that the proposed method not only achieves a good MOS score in general but also generates speech samples with variation, including the style transfer.

Overall, the paper is fairly well organized with figures and tables in appropriate positions. However, it is sometimes difficult to follow, sections 3.3.2 - 3.5.2 in particular, because there are too many short subsections without supporting figures or tables. Furthermore, it is also hard to navigate through the folders and files provided in the supplementary material to listen to the speech samples.

Evaluation is well done. The proposed method yields a comparable MOS score to a reference (Tacotron 2) in terms of speech quality. However, focus in experiments is on the many features that are not feasible in other approaches, which include variation control such as pitch or duration, interpolation between samples, and style transfer. But it still remains unclear that how a user can explicitly control these speech features or attributes in order to generate speech with the desired non-textual information, which requires much more than simple and random variation in pitch and/or duration.

Minor comments:
- There are several typos (even in the abstract) and grammatical errors throughout the text.
- Again, it is quite difficult to find and compare the speech samples that correspond to features describe in the text. It would be better to embed the samples on the website for easier navigation.

---

> ### Author Response · Authors · 2020-11-19
> **Response to AnonReviewer2 11/18/2020**
>
> Thank you for the question you have raised regarding how a user can explicitly control speech features or attributes in order to generate speech with the desired non-textual information.  We have added a new subsection, 3.5 Interpolation between styles (Prior and Posterior), in which we describe how a user can control the speaking style at inference time by simply selecting a sample with the target style and then performing posterior inference to sample from the region in latent space correlated with the target style. In the paper we show figures that illustrate a gradual transition in spectral profile from one style to another. Please listen to our new samples as well to aurally perceive the control over speaking style. We hope this new section clarifies the question you raised.
>
> Generally speaking, the main two points in our paper are representation learning (Normalizing Flows learn an invertible map from data space to a latent space) and Bayesian inference (posterior sampling).  The mapping from data to latent space is such that acoustic properties present in data space will be correlated with regions in the latent space. By sampling from these regions in the latent space we can produce speech with the acoustic characteristics correlated with it.
>
> After finding sample(s) with the speech characteristics of interest, we use Bayesian inference to find and sample from regions in the latent space that contain such speech characteristics. Lambda described in Equation 9 is the parameter we use to navigate between regions in the latent space.
>
> Regarding embedding the samples on the web page, we have updated the web page to include all samples mentioned in the paper and extra samples. We hope it is now easier for you and other readers to navigate and compare our samples.

---

### Official Review · AnonReviewer1 · 2020-10-29
**Very interesting paper that I found rather hard to follow.**

**Rating:** 6
**Confidence:** 3

**Review:**

This paper describes a specific approach to the hot topic of  auto-regressive Flow-based speech synthesis. The fundamentals of coupled flow layers are exposed pretty well, but I quickly got lost in the details of the work. I have a number of specific comments to address that. Nonetheless, I find the paper is still a good (if confusing) read addressing a very interesting area.

"Their assumption is that variable-length embeddings are not robust to text and speaker perturbations, which we show not to be the case" -- in what specific aspects do those works you refer make that assumption?

Re: Figure 1, "Figure 1 shows that acoustic characteristics like timbre and F0 correlate with portions of the z-space of Flowtron models trained without speaker embeddings." Label the axes of Figure 1?

Eq. (7): I can guess, but can you confirm the meaning of the circle-with-dot operator?

"Normalizing flows are typically constructed using coupling layers": I nearly, but don't completely, understand the concept of the coupling layers. AIUI, the trick is that they are forward layers with two components that are inverses of each other.  I guess that Eqs. 5-7 in fact define just that. However, how is z_t computed? f(z_t) and its inverse defined in terms of s_t and b_t, which are functions of z_{1:t-1}. So where does z_t come from?

"we define z1 as a prependded 0 vector" : (only one "d" in prepended). For z1, what is the 0 vector component prepended to? Or is z1 itself a zero vector?

"To evaluate the likelihood, we take the mel-spectrograms and pass them through the inverse steps of
flow conditioned on the text and optional speaker ids, adding the corresponding log |s| penalties, and
evaluate the result based on the Gaussian likelihoods."

For clarity, I suggest:
- Define s.
- Define the complete log-likelihood criterion.

"Our text encoder modifies Tacotron’s 2": spelling.

"Figure 1 shows evidence that speech several characteristics present in mel-spectrograms": check grammar.

"Section 2.5, Posterior Inference": I suggest stating at this point how posterior inference will be used to generate actual samples.

Section 3.2: The inference method has not yet been explained.

Table 1: what happens if you use more Flows? Do the MOS scores improve?

"3.3.1 SPEECH VARIATION": i agree that FlowTron seems to be able to generate more varied speech, but this is only a good thing if the samples are still natural. I don't see a metric or assessment of the naturalness of those more-varied samples (maybe i missed it).

Section 3.3.2, "Our different speaker interpolation samples show that Flowtron is able to gradually
and smoothly morph one voice into another." Where is this measured/reported in the paper?

"3.4.2 SEEN SPEAKER WITH UNSEEN STYLE... We compare samples generated with Flowtron and Tacotron 2 GST to evaluate their ability to emulate a speaking style unseen during training of a speaker seen during training.": where is this comparison shown in the paper?


Table 2: what are the units?

"Table 2 shows that while the samples generated with Tacotron 2 GST are not able to emulate
the high-pitched style from RAVDESS, Flowtron is able to make Sally sound high-pitched as in the
“surprised" style." I find it a bit hard to interpret Table 2. Is the goal to match the reference mean and standard deviation as closely possible?

"3.5 SAMPLING THE GAUSSIAN MIXTURE In this last section we provide visualizations ...": where do you provide it?

re: 3.5.1 VISUALIZING ASSIGNMENTS and 3.5.2 TRANSLATING DIMENSIONS, same comment as before, where are the results provided?



References:
"Sercan O Arik, Mike Chrzanowski, Adam Coates, Gregory Diamos, Andrew Gibiansky, Yongguo
Kang, Xian Li, John Miller, Andrew Ng, Jonathan Raiman, et al. Deep voice: Real-time neural
text-to-speech. arXiv preprint arXiv:1702.07825, 2017b." --> remove superfluous "O" for first author? Or add missing "O" in the previous reference?

---

> ### Author Response · Authors · 2020-11-18
> **Response to AnonReviewer1 11/17/2020**
>
> Thank you for the questions you have raised regarding our paper. We appreciate it that you found our paper to be a good read and hope that our recent modifications based on your comments makes the paper more clear to readers.
> Please take a look at the modifications in the paper and the clarifications below and let us know what remains to be addressed.
>
> Regarding assumptions made by models that use fixed-length embeddings:
> It’s stated in "Towards End-to-End prosody" by R. J. Skerry-Ryan et al. that in their experiments “variable-length prosody embeddings are able to generalize to very long utterances; however, compared to fixed-length embeddings, variable-length embeddings are not as robust to text and speaker perturbations likely because they encode a stronger timing signal. Therefore, this paper focuses on fixed-length embeddings.”.
> In addition, the same paper by R. J. Skerry-Ryan et al also states that “The use of a fixed-length prosody embedding poses an obvious scaling bottleneck, preventing the extension of this approach to longer utterances.”
>
> Regarding dimensions in Figure 1:
> We have updated the paper to clarify that these are TSNE Dimensions.
>
> Regarding understanding coupling layers:
> Please take a look at "NICE: Non-Linear Independent Component Estimation" by Dinh. et al. for a thorough explanation of coupling layers.
>
> Regarding where does z_t come from and what is the 0-vector:
> NN(z_{1:t-1}, text, speaker) will provide scaling and bias terms to affine transform z_{2:t}.
> We use a 0-vector to obtain scaling and bias terms to affine transform z_1.
> Please look at the updated text in 2.2 and let us know if it is clearer now.
>
> Regarding definition of s and the complete log-likelihood criterion:
> In Eq. 5 we define s as the scaling term that is produced by NN. In Sec. 2.1 we define the latent distributions used in our paper.
> When z ~ N(z; 0, I), the log-likelihood log p_{\theta}(z) is equivalent to computing the likelihood of a spherical gaussian.
>
> Regarding "stating at this point ... how posterior inference will be used to generate actual samples.":
> In the original paper this information was provided in Appendix A.2, wherein we provide details on how posterior inference is used to generate samples, while Algorithm 1 describes an algorithm for posterior inference.
>
> Regarding "The inference method has not yet been explained."
> The inference method is defined in "Section 2.4 Inference", right before 2.5. Can you please clarify?
>
> Regarding what happens if you use more Flows:
> Please look at Appendix 1.4.1 where we evaluate the effect of using 2, 3 and 6 steps of flow.
> Through internal evaluations we decided to only compute MOS on models with 2 steps of flow given that they sounded similar to models with more steps of flow while being considerably faster.
>
> Regarding metric for assessment of the naturalness:
> The Mean Opinion Scores implicitly include variety and naturalness given that the Flowtron samples used for collecting MOS are computed with Flowtron using sigma = 0.5, which has more variance than Tacotron 2. Please look at Figure 4 and compare samples from Flowtron sigma = 1.0 with samples from Tacotron 2.
>
> Regarding quantitative measures for interpolation samples and seen speaker with unseen style:
> These are qualitative results. Please listen and compare the samples from each model.
>
> Regarding Table 2 and interpretation:
> We’ve edited the text to emphasize that the goal is to match the pitch summary statistics from the reference providing the style as closely as possible. We've also described that the unit for pitch is MIDI number.
>
> Regarding Gaussian Mixture Visualizations
> Visualizations are provided in the Appendix. We added information to the text to clarify.
>
> Regarding references:
> We updated the references.

---

### Official Review · AnonReviewer4 · 2020-11-03
**FLOWTRON**

**Rating:** 6
**Confidence:** 5

**Review:**

FLOWTRON

1.
The authors present Flowttron, an autoregressive text-to-speech network that is
a merging of the well-established Tacotron architecture along with flow-based
generative models. The benefits to Flowtron are its simplicity to train and
comparable MOS to other state of the art models. Moreover, the model is able to
produce various styles of speech depending on how the latent variables are
sampled, as well style transfer between seen and unseen speakers.

2.
The introduction and discussion in the paper are strong and review the
necessary background information. The experiments are comprehensive and
convincing. Although there are artifacts in the audio samples (repeated
phrases, distortion), the variation of speech is impressive.

The overall description of the model lacks technical thoroughness. The
description of the flow networks is sufficient. However, how precisely
tacotron2 is modified to incorporate the flow network is severely lacking.
Figure 2 is confusing. I also don't understand what the authors mean by "it is
also possible to reverse the ordering of the mel-spectrogram frames in time
without loss of generality"

It is not clear to me how the values in Table 2 are calculated.

On page 4, the second paragraph, what is the purpose of synthesizing samples
with Tacotron2 while varying the pre-net dropout probabilities.

3/4.
I would recommend an accept, given the authors can address the major technical
problems with the paper. Specifically, the description of the system needs to
be much more clear. The work is interesting, the experiments are sufficient,
and the audio samples are convincing.


6. There are many typos that should be fixed.

---

> ### Author Response · Authors · 2020-11-18
> **Response to AnonReviewer4 11/17/2020**
>
> Thank you for the points you have raised in your review. Also thank you for your kind words regarding our work, experiments and samples. We modified our paper to address them. Please take a look at the modifications in the paper and the clarifications below and let us know what remains to be addressed.
>
> Regarding "how precisely Tacotron 2 is modified to incorporate the flow network":
> Specifically, in addition to comparing Tacotron 2 and Flowtron in the body of the paper, we have provided images (Figure 2 for Flowtron and Figure 6 in the Appendix A.4.4 for Tacotron 2) that can be used to compare Flowtron's architecture with Tacotron’s architecture. Finally we’ve also provided model summaries for Flowtron and Tacotron 2 in Appendix A7.
> Regarding what is mean by "it is also possible to reverse the ordering of the mel-spectrogram frames"
> In most TTS models, including Tacotron, mel-spectrogram generation is done from past frames to future frames.
> In Flowtron, by reversing the ordering of mel-spectrogram frames on certain steps of flow, we allow the model to also generate mel-frames in the reverse order. We have observed, and described it in A.4.2 that bidirectional processing improves speech quality when compared to unidirectional processing.
>
> Regarding how Table 2 is calculated:
> Table 2 provides pitch summary statistics computed over 5 phrases and 10 takes each. The goal is to produce samples whose pitch summary statistics (mean and standard deviation) are closer to the reference. Please let us know if the new caption for Table 2 makes the point clearer.
>
> Regarding the purpose of synthesizing samples with Tacotron 2 while varying the pre-net dropout probabilities:
> In the Tacotron 2 paper, the authors state that “In order to introduce output variation at inference time, dropout with probability 0.5 is applied only to layers in the pre-net of the autoregressive decoder”. Increasing or decreasing the dropout probability during inference can be seen as a mechanism to increase or decrease variation in Tacotron 2 at inference time.

---

### Decision · Program_Chairs · 2021-01-07
**Final Decision**

**Decision:**

Accept (Poster)

**Comment:**

This paper proposes an autoregressive flow-based network, Flowtron, for TTS with style transfer. It integrates the Tacotron architecture with the flow-based generative model.  Extensive experiments are carried out in a controlled manner and the results show that the proposed Flowtron framework can achieve comparable MOS scores to the SOTA TTS models and is good at generating speech with different styles. All reviewers consider the work interesting.  There are concerns raised on technical details which mostly have been cleared by the authors' rebuttal. The exposition also has been greatly improved based on the reviewers' suggestions and questions.  Overall, this is an interesting paper and I would recommend acceptance.